# *Commiphora wildii* Merxm. Essential Oil: Natural Heptane Source and Co-Product Valorization

**DOI:** 10.3390/molecules28020891

**Published:** 2023-01-16

**Authors:** Djallel Mansouri, Anne Landreau, Thomas Michel, Clément De Saint Jores, Bienvenue Razafimandimby, Marie Kempf, Stéphane Azoulay, Nicolas Papaiconomou, Xavier Fernandez

**Affiliations:** 1Institut de Chimie de Nice, Université Côte d’Azur, CNRS UMR 7272, F-06108 Nice, France; 2Infections Respirations Fongiques, Interactions Cellulaires et Applications Thérapeutiques 2, SFR 4208, Université d’Angers, Université de Brest, F-49000 Angers, France; 3Laboratory of Bacteriology, University Hospital, F-49045 Angers, France; 4Immunologie et Nouveaux Concepts en Immunothérapie, INSERM, CHU Angers, Université d’Angers, Nantes Université, F-49000 Angers, France

**Keywords:** *Commiphora wildii*, essential oil, heptane, co-product valorization, *C. glabrata*, *C. albicans*

## Abstract

As an alternative to fossil volatile hydrocarbon solvents used nowadays in perfumery, investigation on essential oil of *Commiphora wildii* Merxm. oleo gum resin as a source of heptane is reported here. Heptane, representing up to 30 wt-% of this oleo gum resin, was successfully isolated from the *C. wildii* essential oil, using an innovative double distillation process. Isolated heptane was then used as a solvent in order to extract some noble plants of perfumery. It was found that extracts obtained with this solvent were more promising in terms of sensory analysis than those obtained from fossil-based heptane. In addition, in order to valorize the essential oil depleted from heptane, chemical composition of this oil was found to obtain, and potential biological activity properties were studied. A total of 172 different compounds were identified by GC-MS in the remaining oil. In vitro tests—including hyaluronidase, tyrosinase, antioxidant, elastase and lipoxygenase, as well as inhibitory tests against two yeasts and 21 bacterial strains commonly found on the skin—were carried out. Overall, bioassays results suggest this heptane-depleted essential oil is a promising active ingredient for cosmetic applications.

## 1. Introduction

Because of the large number of volatile compounds exhibiting pleasant odor, flavor and biological activities present in essential oils, the latter are valorized in various ways, such as aromas, perfumes and cosmetics, as well as pharmaceutical drugs [1,2]. When a compound inside an essential oil is relatively abundant, it can be physically isolated (e.g., fractional distillation). For instance, Chemat et al. reported the extraction of limonene from citrus peels essential oil and its valorization as a green solvent [3,4]. Similarly, menthol was extracted and further purified from the essential oil of *Mentha arvensis* by fractional distillation [5].

*Commiphora wildii* Merxm. is a Namibian desert resin tree. Its aromatic exudate is traditionally used as a perfume by women from the Ovahimba tribe, originating from Namibia. The resin is also used in combination with animal fat and ochre to protect skin against solar exposure [6]. Up to now, *C. wildii* essential oil has received little attention leading to limited knowledge about its phytochemical content, compared to studies reported for *C. myrrha* [7,8,9,10]. Very recently, two works described the terpenoid content of *C. wildii* [6,11]. The essential oil of this Namibian resin analyzed by Sheehama et al. [6] shows a surprisingly high level of heptane (24%) along with *α*-Pinene (50.0%) and *β*-Pinene (11.7%). Jemmali et al. [11] have characterized monoterpenoids, diterpenoids and triterpenoids in *C. wildii* resin after solvent extraction such as chloroform, and sample derivatization. Other species, such as *Pinus oocarpa*, *P. jeffreyi* and *P. sabiniana* contain large amounts of heptane in their essential oil obtained from exudates or from woods [12,13]. But to the best of our knowledge *C. wildii* is the only one in the *Commiphora* genus presenting such a high amount of natural heptane. *C. wildii* can thus be envisaged as a viable source of sustainable heptane to be applied as an extraction solvent in perfume industry.

There are three goals for this study: (i) to describe isolation of heptane from *C. wildii* essential oil; (ii) to evaluate sustainable heptane as an extraction solvent of raw materials for perfumery; and (iii) to study the chemical composition of the remaining essential oil, after heptane isolation, and its possible bioactivities. The latter goal could provide insights into the reasons for the traditional use of this plant and give possible ways of valorizing this extract. Such results have led to a patent recently filed by our research group [14].

## 2. Results

### 2.1. Chemical Composition of Essential Oil of C. wildii and Heptane Isolation

Isolation of heptane from *C. wildii* was achieved by using an innovative two-step process, so-called “double-distillation” process. The first step consists in a classical hydrodistillation from *C. wildii* resin, leading to the production of an essential oil (EO). In the second step, a fractional distillation of the essential oil leads to one enriched heptane fraction and one oily fraction corresponding to the remaining essential oil. In the essential oil from *C. wildii* resin obtained after the first step, 58 compounds were identified (98.5% of the total GC area; Appendix A). Compounds were mainly monoterpene and sesquiterpene hydrocarbons, along with alkane. Oxygenated terpenes were also characterized. As shown in Table 1, most abundant compounds were *α*-pinene (43.4%), heptane (29.5%) and *β*-pinene (11%), in accordance with previous work [6].

The proportion of heptane was determined by external calibration (protocol described in materials and methods section) and was found to be around 25 to 35% in mass, depending on the batch from the supplier. These results are in agreement with the one reported by Sheehama et al. (24 ± 7%) [6].

During the second step of the process, a fraction of around 35 wt% was separated from the rest of the essential oil by fractional vacuum-distillation. This fraction contained heptane (>99%), *α*-pinene (0.3%) and *β*-pinene (0.1%).

### 2.2. Heptane from C. wildii as An Extraction Solvent for Perfumery

Heptane obtained was evaluated as solvent for extracting three emblematic and historical flowers of French perfumery: rose (*Rosa centiflolia* L.), Madone lily (*Lilium candidum* L.) and jasmine (*Jasminum grandiflorum* L.). Both extractions were carried out under the same conditions. Exception made from two peaks was found on the chromatogram of concrete obtained using heptane from *C. wildii*, GC-MS analyses of each concrete led to similar chromatograms within experimental uncertainty. These two peaks were attributed to *α*-pinene and *β*-pinene, which are removed from the essential oil together with heptane during the double distillation process. Olfactory comparison (use of a panel trained in odor description) between concretes extracted with fossil heptane and those obtained from natural heptane was performed.

Results, collected in Table 2, reveal that different olfactory perceptions between concretes were obtained using heptane from *C. wildii* EO and those obtained using fossil heptane. Yields obtained using natural heptane appears to be higher than those obtained using fossil heptane, and to exhibit slightly different olfactive profiles, with a generally more floral note. Olfactive description was carried out with the help of an independent perfumist.

Even if natural heptane extracted from this essential oil cannot be used in large quantities, it offers a natural alternative to petrochemical hydrocarbons, especially to extract noble fragrances from plants (i.e., concrete) [15] in limited productions.

### 2.3. C. wildii Essential Oil after Heptane Isolation

The remaining essential oil, after heptane isolation, represents nearly 70% of starting essential oil. This represents an important quantity that must be valorized.

The essential oil composition of *C. wildii* resin, after heptane isolation, was studied by GC-MS and GC/FID (Figure 1). To achieve this, the residual essential oil was analyzed directly and after fractionation by silica gel chromatography (Appendix A, Appendix A). Compounds identified are presented in Table 3 along with their elution time. Chromatographic profiles of the essential oil before fractionation revealed 97 compounds. After fractionation by silica gel chromatography, 172 compounds were identified, representing 98.36% of total GC/FID area. This number is much larger than those previously identified by Sheehama et al. [6]. Qualitatively, mainly oxygenated monoterpenes (72), monoterpenes (24), sesquiterpenes (14), oxygenated sesquiterpenes (10), diterpenes (2), hemiterpene (1) but also acids (5), alkanes (6), alcohols (9), aldehydes (3), ketones (13), esters (6), phenols (3), aromatic hydrocarbon (1) and furans (3) were found.

The most abundant chemical group is monoterpenes (94.04% of the FID total area), followed by alkanes (2.84%) and oxygenated monoterpenes (1.47%).

The ten major compounds of the residual essential oil are *α*-pinene (63.55%), *β*-pinene (15.95%), sabinene (5.01%), *α*-thujene (4.21%), heptane (2.31%), *p*-cymene (2.10%), limonene (1.22%), camphene (0.68%), terpinene-4-ol (0.64%) and nonane (0.53%).

### 2.4. Bioactivities of the Remaining Essential Oil

Because of the amount of compounds found in residual *C. wildii* essential oil after heptane isolation and in order to find potential applications of such an essential oil, cosmetic potential and antimicrobial potential was studied. To that end, several bioactivity assays, such as hyaluronidase, tyrosinase, antioxidant, elastase and lipoxygenase assays [16] were carried out. Results are presented in Figure 2. Finally, activity of such an essential oil on several fungi and bacterial strains was measured. Results are presented in Table 4 and Table 5.

#### 2.4.1. Bioassays Results

Bioassay results obtained for doubly distilled *C. wildii* essential oil are presented in Figure 2. These results show that this essential oil exhibits significant hyaluronidase and lipoxygenase activities but no tyrosinase nor antioxidant activities. Finally, elastase activity is low compared to positive response since its inhibition reaches 22% compared to 72% obtained for the positive response.

#### 2.4.2. Antimicrobial Tests

Antifungal activity of the doubly distillated essential oil was tested on two current yeasts and according to two different methods, namely agar diffusion and broth dilution method. Inhibition results are presented in Table 4.

Essential oil of *C. wildii* exhibited an inhibitory effect against both species of *Candida albicans* and *glatrata*, the inhibition diameter for the latter being slightly larger than that obtained on *C. albicans*. Nevertheless, the same MICs values were obtained. It can be noticed that essential oils from other species of the genus *Commiphora* seem to have a similar behavior against *C. albicans* [17,18,19].

On the other hand, the antibacterial activity of *C. wildii* essential oil has also been evaluated against various bacterial strains capable of colonizing mainly the oral cavity or skin (Table 5).

An interesting activity from 30 mg/mL on all strains of Methicillin-resistant *S. aureus* and *S. pyogenes* was observed. An inhibition is also detected on *C. tuberculostearicum* and *P. asaccharolytica* strains from a value of 100 mg/mL. However, no activity was found at the concentration tested on all other bacteria.

## 3. Discussion

Because of the amount of heptane found in the essential oil of *C. wildii*, typically 30%, it is conceivable to use this renewable source of heptane as an alternative source of natural solvent for the extraction of very valuable raw materials for perfumery. Extraction of three noble plants, as shown in Table 2, reveals that concrete yields are generally higher when heptane from *C. wildii* is used compared to fossil heptane. This is most probably due to the presence of odorous compounds, such as 0.3% *α*-pinene and 0.1% *β*-pinene, that are present in this heptane and that, because of a high volatility of these compounds, remain with heptane throughout the double distillation process. Once extraction from a plant such as *Rosa centifolia* or *Jasminium grandifolium* is carried out using this heptane, *α*-pinene and *β*-pinene, and possibly other molecules present in smaller amounts in heptane, remain within concrete after heptane removal, thus leading to a yield higher than expected.

Moreover, slight olfactory note variations are observed as well. These variations, once again, are most probably due to the presence of *α*-pinene and *β*-pinene remaining in concrete after removal of heptane from *C. wildii*., thus modifying slightly the olfactive profile of the concrete, compared to that obtained using fossil heptane.

To the best of our knowledge, except for antioxidant activity, biological activities of *C. wildii* essential oil, were not reported previously [6]. Hyaluronidase is an enzyme responsible for depolymerization of hyaluronic acid [20,21]. This acid, with its distinctive feature of retaining 6 L of water in 1 g of hyaluronic acid [22,23], ensures the hydration and softness of the skin, promotes wound healing and the reduction of wrinkles [21,22,23]. Degradation of hyaluronic acid by hyaluronidase causes the viscosity of body fluids to decrease and the permeability of connective tissues to increase [21,24]. Thus, hyaluronidase inhibitors, by their effective regulatory factors, preserve moisture and skin smoothness, by regulating and balancing both catabolism and anabolism processes [23]. Our experiments showed that the essential oil demonstrated an inhibitory activity of 86% in the hyaluronidase activity, a value greater than that of hydraberry commercial extract, that was used as a control (71% inhibition) for this test. This significantly high inhibitory effect might explain traditional use of *C. wildii* as a cosmetic in order to restore the skin.

Melanogenesis is the process of production of the melanin by cells called melanocytes [25]. Melanin is a main determinant of skin color, and it provides protection to the skin by absorbing 50% to 75% of ultraviolet rays. Also, these pigments have the ability to scavenge reactive oxygen species (ROS) [26,27]. In melanocytes, melanin synthesis is catalyzed by Tyrosinase. This enzyme is responsible for the hydroxylation of L-tyrosine into 3,4-dihydroxyphenylalanine (L-DOPA) and the subsequence oxidation of L-DOPA to dopaquinone. The latter is a highly reactive compound able to polymerize spontaneously to build melanin. In any case, studies [23,28] have concluded that in the skin, an excessive production of melanin can lead to melanoma and hyperpigmentation and can be genotoxic. However, the European commission restricts the use of a large list of synthetic molecules despite their strong bleaching ability [29,30]. Subsequently, in the field of cosmetics and pharmaceuticals, tyrosinase inhibitors have become molecules of great importance as lightening agents [23]. Regarding lightening activity, the essential oil tested here shows a low inhibitory activity (3%), much lower than that of SymWhite control (100%). Our results are in agreement with those commonly obtained for plant extracts [31,32,33,34].

*C. wildii* essential oil exhibits an anti-elastase activity of 22%, a value significantly lower than that of Berryflux vita control, which exhibits a value of 72,42%. Previous work reported an inhibition activity of *α*-pinene and limonene on elastase [35]. Because *α*-pinene is the major compound of *C. wildii* essential oil after heptane isolation, and because limonene is also among the 10 most abundant compounds found in this essential oil, they are probably responsible for this non negligible anti-elastase activity.

The lipoxygenase activity for this essential oil is high, reaching a typical value of 96%. This value is slightly higher than that of Resveratrol control. Recent studies [36,37,38,39] report that *α*-pinene has an anti-inflammatory effect. The presence of this compound in large proportion in the essential oil obtained in this study most probably explains the high lipoxygenase activity observed for *C. wildii* essential oil.

Starting for the results obtained here, it appears that *C. wildii* essential oil obtained after the double distillation process has a cosmetic potential, such as an active ingredient in the development of anti-aging or skin repairing cosmetic active.

*Cutibacterium acnes* belongs to largely commensal species and is an integral part of the cutaneous flora present on the skin of most healthy adult humans [40]. Species of the genus *Prevotella* sp. are part of the oral, vaginal and intestinal microbiota and are often found after an anaerobic infection of the respiratory tract. *Gemella morbillorum* is rarely a cause of disease in humans, though it may be found benignly in the oropharyngeal area. It has been reported to be among the most common bacteria present in teeth with cysts that do not resolve after repeated treatments [41]. *Porphyromonas sp*. is a pathogenic bacterium that causes periodontal disease. This genus lives in the oral cavity of man and is part of the salivary microbiome. Finally, *Streptococcus pyogenes* and *Staphylococcus aureus* are commensal bacteria responsible for potentially serious infections in humans. It appears that the essential oil of *C. wildii* shows preferentially an activity on pathogenic and not commensal strains of the skin or oral cavity.

Sheehama et al. (2018) [6] measured the antibacterial activity of the essential oil of *C. wildii* in vitro against three bacteria (*E. coli*, *K. pseudomoniae* and *S. aureus*) and one fungi (*C. albicans*). The MICs measured were 10 mg/mL for all strains except for *S. aureus* was the best antibacterial activity measured with an MIC 8 mg/mL. This result for *C. albicans* is greater than that obtained in our experiment. It is also important to note that MIC (Minimum inhibitory concentration) obtained in our study for antibacterial activity is at least three times higher than that obtained by Sheehama et al.

A recent report [42] showed that (+)-*α*-pinene and (+)-*β*-pinene exhibited a high toxicity against *C. albicans*, 100% of the inoculum killed in 60 min. The presence of these compounds in large proportion in the residual essential oil reported in this study could explain the results observed with *C. albicans*. [42].

In conclusion, the essential oil of *C. wildii* obtained after the double distillation process has shown biological activities such as hyaluronidase and lipoxygenase inhibition, antifungal activities against *C. albicans* and *C. glabrata* and antibacterial activity against some strains of methicillin-resistant *S. aureus* and *S. pyogenes*. It can be noticed that the two fungi studied here are responsible of dermal infections. Therefore, considering the biological and antimicrobial activities, this product could be used as a raw material in the development of cures or drugs against skin infections without unbalancing the natural microbiota, and for its antiaging and skin repairing properties.

## 4. Materials and Methods

### 4.1. Plant Material, Chemicals and Reagents

Essential oil of *Commiphora wildii* was obtained from BeHave (Nantes, France). As the bark of this tree is very thin, and the resin-secreting channels are located just underneath, it is possible to recover this resin by incision on the low part of the trunk. The exudate used in this study was collected after some weeks so that the resin was dry [14].

Analytical grade solvents (i.e., pentane, diethyl ether, methanol) as well as dimethyl sulfoxide (DMSO), amphotericin B, and (YPDA) agar containing 0.5 g/L of chloramphenicol were purchased from Sigma–Aldrich (Saint Quentin Fallavier, France).

### 4.2. Isolation of Natural Heptane

Bio-sourced heptane was obtained from the commercial essential oil by vacuum distillation. Essential oil (4 kg) was introduced in a 6 L round-bottom flask overcome by a Sulzer^®^ packed column 1 m long with a diameter of 10 cm. The distillation temperature was set at 30 °C; eight fractions were collected then analyzed by GC/MS.

### 4.3. Fractionation of Residual Essential Oil (Silica Gel Chromatography)

Essential oil obtained after heptane isolation (5 g) was fractionated on a silica gel column (50 g). Five fractions were obtained: F1 (300 mL pentane), F2 (200 mL pentane/diethyl ether 90/10 *v*/*v*), F3 (200 mL pentane/diethyl ether 50/50 *v*/*v*), F4 (200 mL diethyl ether) and F5 (100 mL methanol) (Appendix A).

### 4.4. Extraction and Evaluation of Concretes

Isolated heptane was used as a solvent for extracting natural fragrant raw materials. The extractions were carried out under the same conditions for three plants, obtained from the Jardin du Musée International de la Parfumerie (Mouans-Sartoux), (*Rosa centifolia* L., *Lilium candidum* L., *Jasminum grandiflorum* L.). Frozen plant material (5 g) was extracted in 50 mL of heptane, natural or fossil. After 4 h of stirring, a filtration of the macerate on a Büchner then an evaporation of the heptane from the filtrate in a rotary evaporator were carried out in order to obtain a concrete.

Concrete was then washed with 2 mL of ethanol and then placed in the freezer for 2 h. The solidification of waxes was provided by the glassing step; then filtration separated them from the rest of the extract. Absolute was obtained after final evaporation of the filtrate.

An olfactory comparison was made between the extracts obtained and those derived from fossil heptane, from a petrochemical source. These olfactory evaluations were performed by trained perfumers.

### 4.5. GC-MS and GC/FID Analyses of Complete and Residual Essential Oil

Analysis of the residual essential oil was performed by GC-MS and GC/FID using an Agilent 6890N gas chromatograph (Palo Alto, CA) equipped with an Agilent MSD5973N mass selective detector, a flame ionization detector (FID), an electronic pressure control (EPC) injector and a multifunction automatic sampler (Combi-Pal, CTC Analytics, Zwingen, Swiss). Separations were achieved either on an apolar HP-1 capillary column (100% polydimethylpolysiloxane; 50 m × 200 µm, 0.33 µm film thickness, Agilent Technologies) or a polar column HP-Innowax 50 m × 200 µm, 0.4 µm film thickness, Agilent Technologies. 1μL of sample (80 mg/mL) was injected in split mode (1/25), and helium (carrier gas) was used at a flowrate of 0.8 mL/min. The injector temperature was set to 250 °C, and the oven temperature was programmed from 40 °C to 270 °C at 3 °C/min for the apolar column; the oven temperature was programmed from 40 °C for 5 min, then 40 °C to 220 °C at 2 °C/min for the polar column. For GC-MS, a solvent delay of 5 min was selected. Mass spectra were recorded in electronic ionization (EI) mode at 70 eV scanning the *m*/*z* 35–500 range (3.15 scan/s). For GC-FID, samples were injected in triplicate for quantification. The average of these three values and the standard deviation were determined for each identified compound.

*C. wildii* essential oil and the concretes extracted with natural heptane were identified by GC-MS using apolar column (Supelco SLB—5MS; 30 m × 250 µm, 0.33 µm film thickness). 1μL of sample (80 mg/mL) was injected in split mode (1/100), and helium was used at a flowrate of 1 mL/min. The injector temperature was set to 250 °C, and the oven temperature was first programmed from 40 °C to 220 °C at 2 °C/min and then from 220 °C to 270 °C at 20 °C/min.

### 4.6. Identification of Complete and Residual Essential Oil Compounds

Data treatment was performed using MSD ChemStation (E02.02) software, Agilent Technologies. The identification of compounds involved a comparison of mass spectra with those recorded by internal or commercial mass-spectral libraries (Flora, NIST and Wiley), as well as comparison of linear retention indices (LRI) with those available in the literature (NIST, ESO) and articles for data missing from databases mentioned above [2,43,44,45,46,47,48,49,50]. Retention indices (RI) were calculated using a formula according to van Den Dool and Kratz [51] and according to the retention times of standard n-alkanes C6-C27 homemade mixture. Alkanes mixture diluted to 10% in diethyl ether was analyzed by GC-MS and GC/FID, according to the above-described methods.

### 4.7. Heptane Assay in Complete Essential Oil

A heptane assay method has been developed using GC-FID in order to quickly check heptane content in samples. A calibration curve with fossil heptane in hexane at five chosen concentrations (0%; 20%; 40%; 60%; 80% *m*/*m*) was performed with linear regression (Appendix A).

Heptane assays by GC/FID were performed on an Agilent Intuvo 9000 chromatograph with a G4513A autosampler equipped with a flame ionization detector (temperature set at 250 °C, air flow at 300 mL/min, H2 flow at 30 mL/min). The column used is an apolar column HP-5 MS 30 m × 0.25 mm × 0.25 μm, with H2 as carrier gas set at 1 mL/min. A 0.1 µL volume injection volume was used, with a split ratio of 1/100. Temperature program mode was used for the oven: 1 °C/min from 40 °C to 50 °C and then 20 °C/min up to 270 °C, followed by 20 min at 270 °C.

### 4.8. Activity Tests

#### 4.8.1. Bioassays

Bioassays were carried out as presented in previous studies [52,53]. Samples, such as extracts, standards and controls, were prepared at a concentration of 3.433 mg/mL in dimethyl sulfoxide (DMSO). Positive controls used for all bioassays are collected in Table 6. In each plate, a negative control corresponding to neat DMSO (OD control, with OD statin for optical density), exhibited no activity for all bioassays.

##### Instrumentation

Bioassays were conducted in untreated 96-well UV-transparent plates obtained from Costar, Sigma-Aldrich (Saint-Quentin Fallavier, Auvergne-Rhone-Alpes, France) or purchased from Thermo Nunc (Villebon-sur-Yvette, Ile-de-France, France). In order to seal all 96-well plates during incubation, adhesive sealing films (Greiner Bio-One, Courtaboeuf, Ȋle de France, France) were used. Bioassays were conducted using an automated pipetting system epMotion 5075 from Eppendorf. Absorbance of 96-well plates was recorded using a microplate reader Spectramax Plus 384 from Molecular Devices, Wokingham, Berkshire, UK. Data acquisition was provided by SoftMaxPro software (Molecular devices, Wokingham, Berkshire, UK) and the inhibition percentages calculated using Prism software (GraphPad Software, La Jolla, CAS, USA). Unless specified, results are reported as inhibition percentages (I%) calculated as follows for the DPPH radical scavenging assay, and tyrosinase, elastase and lipoxygenase assays:I% = ((OD control − OD sample)/OD control) × 100 (with OD starting for optical density).
Or as follows (for the hyaluronidase assay):
I% = (OD sample/OD blank-OD control) × 100

Likewise, after correction of each OD (except those for hyaluronidase) using the blank measurement related to the sample absorbance value, the substrate was added.

##### Hyaluronidase Assay

Assay was carried out as follows: in each well, 150 µL of a solution of hyaluronidase (13.3 U/mL in hyaluronidase buffer) was deposited along with 7.5 µL of extract. Incubation for the sealed plate was performed at 37 °C for 20 min, and a first OD reading was carried out at 405 nm. Next, in each well, 100 µL of a solution of hyaluronic acid (150 µg/mL in pH 5.35 buffer) was added. 50 µL of cetyltrimethylammonium bromide (40 mM in a 2% NaOH solution), after 30 min incubation at 37 °C, was distributed in each well, and a final OD reading was carried out.

##### Tyrosinase Assays

Assay was carried out as follows: in each well, 150 µL of a solution of mushroom tyrosinase (171.66 U/mL in phosphate buffer) was added along with 7.5 µL of extract. Incubation of the sealed plate was performed at RT for 20 min. Next, in each well, 100 µL of a solution of substrate (either L-tyrosine or L-DOPA, 1 mM in phosphate buffer) was deposited. After 20 min of incubation, the final OD reading was carried out at 480 nm

##### DPPH Radical Scavenging Assay

Evaluation of the antioxidant activity of an extracts was carried out by studying the scavenging activity of 1,1-diphenyl-2-picrylhydrazyl radical (DPPH) (4,5,6). In each well, 150 µL of a solution of ethanol/acetate buffer 0.1 M (50/50) was added, with 7.5 µL of a sample.

At 517 nm, a first OD reading was made (OD blank). Then 100 µL of a DPPH solution (386.25 µM in ethanol) were deposited in each well. After incubation of the sealed plate at RT for 30 min in the dark, a final OD reading was performed.

##### Elastase Assay

In each well, 150 µL of a solution of porcine pancreatic elastase (0.171 U/mL in Tris buffer) was deposited, with 7.5 µL of the extracts. Incubation of the sealed plate was performed at RT for 20 min. After recording a first OD reading at 410 nm, 100 µL of a solution of N-succinyl-Ala-Ala-Ala-p-nitroanilide (2.06 mM in Tris buffer) was added. Then, a 40-min incubation was carried out before final OD reading.

##### Lipoxygenase Assay

In each well, 150 µL of a solution of soybean lipoxygenase (686.66 U/mL in phosphate buffer) was added, along with 7.5 µL of an extract. Incubation of sealed plate was performed for 10 min in the dark. An incubation was carried out in the dark for 2 min, before a first OD reading at 235 nm; after an additional incubation of 50 min, the final OD reading was carried out.

#### 4.8.2. Antimicrobial Activity

##### Antifungal Activity

Antifungal activity of extracts was tested against human pathogenic fungi, two yeasts (*Candida albicans* and *C. glabrata*). These yeasts were obtained from Parasitology and Mycology laboratory of CHU d’Angers (France). They were incubated at 37 °C on yeast extract-peptone-dextrone (YPDA) agar containing 0.5 g/L of chloramphenicol for a period of 48 h. Classically, the extracts or the compounds tested were dissolved in DMSO to obtain an initial concentration of 10 mg/mL. In our case, the essential oil was tested as pure without being diluted in DMSO. The fungal strains tested are listed in Table 4.

###### Antifungal Evaluation


Disk diffusion testing


This method is performed according to a method traditionally used for yeasts [54]. *Candida* suspensions are obtained by incorporating a 2 mm colony in sterile distilled water [corresponding to 3 × 10^6^ CFU per mL (Colony Forming Unit) for *C. albicans*, and 5 × 10^6^ CFU per for *C. glabrata*]. A volume of 25 µL of essential oil was deposited on 12 mm diameter paper disks (Prat Dumas, France). After drying a Petri dish (diameter 90 mm) containing Casitone agar, the disks are placed at the center previously inoculated with 10 mL spore suspensions. Amphotericin B was used as a positive control and DMSO without compounds served as a negative control. Evaluation was done after 48 h of incubation by measurement of size of growth inhibition zones (mm) around the disk papers.

###### Broth Dilution Testing

Tests were carried out according to the recommendations of the CLSI in accordance with the reference method M27-A3 [55] for yeasts. Briefly, after the required culture time, Petri dishes are scraped with 2 × 10 mL of sterile distillated water and then centrifuged. The pellet is then washed, and suspensions are spectrophotometrically adjusted at 630 nm to a final concentration of approximately 0.5 × 10^3^ to 2.5 × 10^3^ CFU per mL for the yeasts. The suspensions are prepared in RPMI-1640 medium supplemented with 2 L-glutamine and buffered to 0.165 M with MPSO (3-(N-morpholino) propanesulfonic acid). Tests are carried out in sterile 96-well plates. From the essential oil tested, serial dilutions to half are made in DMSO. The solutions prepared are distributed in triplicate in the wells at a rate of 5 µL in final volume of 200 µL. Amphoterecin B is used as positive control. The growth control is realized in triplicate, using the spore suspension supplemented with 5 µL of DMSO without compounds.

After 48 h at 37 °C for *C. albicans* and *C. glabrata*, the minimum inhibitory concentration (MIC) is determined via solution turbidity as being the minimum concentration causing an inhibition equal to or greater than 80% of the inhibition caused by growth control.

For DMSO, the final concentration is less than 2.5%, which does not significantly affect fungal growth.

###### Antibacterial Activity

Antibacterial activity was evaluated on 21 clinical isolates collected by the Laboratory of Bacteriology at the University Hospital of Angers, France. The bacterial strains tested were: five *Cutibacterium acnes*, three *Prevotella buccae*, two methicillin-resistant *Staphylococcus aureus*, three methicillin-susceptible, two *Corynebacterium tuberculostearicum*, one *Gemella morbillorum*, one *Gemella haemolysans*, one *Porphyromonas asaccharolytica* and three *Streptococcus pyogenes*.

Tests were performed using a modified methodology described by Alomar et al. and adapted to anaerobic bacterial strains [56].

Briefly, a stock solution of *C. wildii* essential oil was prepared at 4 g/mL in DMSO under sterile conditions. The concentrations tested were 3, 30, 100 mg/mL, and tests were performed in Petri plates in a final volume of 20 mL Mueller-Hinton agar (Merck, Darmstadt, Germany) and 1 mL of horse serum.

####### Preparation of Bacterial Inocula

For each bacterial isolate, a concentration corresponding to a 0.5 MF was prepared in sterile physiological serum, using a densitometer. This suspension corresponds approximately to 10^8^ bacteria/mL. Then, 4 µL of each suspension were inoculated using an automatic inoculator (multipoint inoculator AQS), on the Petri dishes containing the MH agar with the three concentrations of *C. wildii* extract to be tested. Control Petri dishes without *C. wildii* essential oil were also used as growth control. After incubation for 24 h at 37 °C under anaerobic conditions, the minimum inhibitory concentration (MICs mg/mL) of *C. wildii* against each bacterial strain was determined. The MIC corresponds to the lowest concentration leading to bacterial growth inhibition. The experiment was carried out in duplicate.

## 5. Conclusions

For the first time, *C. wildii* was used as a source of heptane naturally present in its essential oil. The latter was found to contain up to 30% heptane. A simple double distillation process was proposed, yielding heptane of high purity, containing 0.3% *α*-pinene and 0.1% *β*-pinene. Heptane was used as an extraction solvent in order to obtain concrete from three emblematic and historical flowers of french perfumery. Because of the olfactive impurities present in heptane isolated from *C. wildii* essential oil and remaining after evaporation of heptane, yields for concretes obtained from such natural heptane were found to be higher than those obtained using fossil heptane. Furthermore, olfactive notes for concretes obtained using the so-called natural heptane were slightly different, in particular somehow more floral, compared to those obtained using fossil heptane, offering a new potential range of raw material for perfumer.

In addition, investigation on the composition of the remaining essential oil after heptane isolation revealed ten compounds were found in majority in the essential oil, *α*-pinene being the most abundant compound.

Finally, resulting essential oil after removal of heptane exhibited important hyaluronidase and lipoxygenase activities but low elastase activity. These activities suggest possible interesting antiaging and skin repairing cosmetics properties. These properties seem to conform to the traditional usage of the resin of *C. wildii*.

Tests against several strains of bacteria and two *Candida* species revealed inhibition preferentially on pathogenic and not commensal human bacterial strains of the skin or oral cavity, a medium effect against *C. albicans* and a potent inhibitor of *C. glabrata*. This study has shown different biological properties for the doubly distillated essential oil of *C. wildii*, which could be valorized as a raw material for cosmetic usage, after the removal of the natural heptane for perfumery usage.

The overall small amount of heptane extractable from *C. wildii* implies necessarily its application to luxury perfumery or other small scale production units.

## 6. Patents

Bouville, A.-S.; Dieffoldo, C.; Fernandez, X.; Piquart, S. Heptane from a Plant Source, for the Extraction of Natural Products; 9 December 2020, EP3746038.

## Figures and Tables

**Figure 1 molecules-28-00891-f001:**
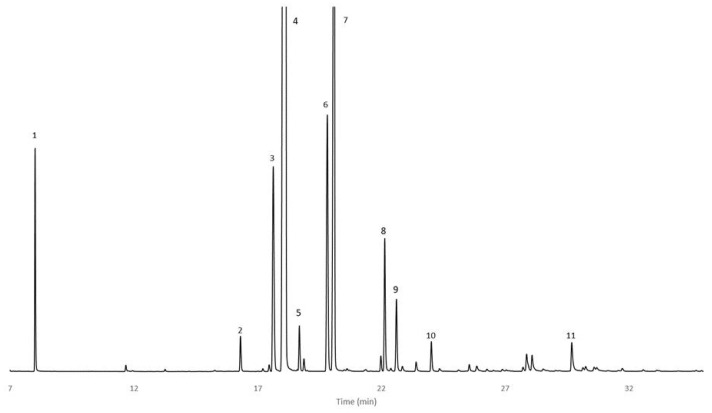
GC/FID chromatogram of *C. wildii* doubly distillated essential oil from 7 to 35 min: **1** Heptane, **2** Nonane, **3**
*α*-Thujene, **4**
*α*-Pinene, **5** Camphene, **6** Sabinene, **7**
*β*-Pinene, **8**
*p*-Cymene, **9** Limonene, **10**
*γ*-Terpinene, **11** Terpinene-4-ol (conditions in Section 4).

**Figure 2 molecules-28-00891-f002:**
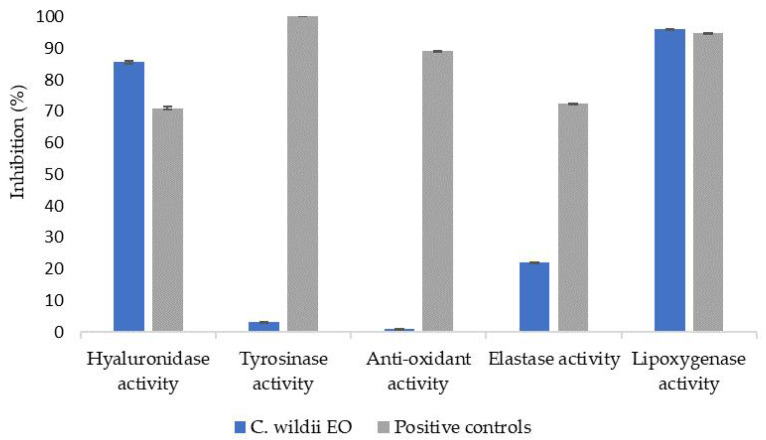
Bioactivity tested for essential oil of *C. wildii* resin after heptane isolation, compared with bioactivity of commercial cosmetic ingredients (positive controls: hyaluronidase: commercial hydraberry extract, tyrosinase: SymWhite ingredient, Anti-oxidant: commercial rosemary extract, elastase: Berryflux vita (VITALAB), lipoxygenase: resveratrol).

**Table 1 molecules-28-00891-t001:** Major compounds of *C. wildii* essential oil (GC/MS) before heptane isolation (compounds with relative percentage greater than 0.5%).

Compounds	Experimental RI	Theorical RI	%
Heptane	702	700	29.5
Nonane	900	900	0.9
*α*-Thujene	924	930	3.7
*α*-Pinene	934	932	43.4
Camphene	946	955	0.7
Sabinene	970	969	0.5
*β*-Pinene	976	974	11.0
o-Cymene	1023	1023	2.3
Limonene	1027	1024	1.0
*γ*-Terpinene	1056	1062	0.7

**Table 2 molecules-28-00891-t002:** Comparison of olfactive evaluation obtained for the extraction of tree natural products, using fossil heptane and heptane from *C. wildii*.

Species	Heptane from *C. wildii*		Fossil Heptane	
	Yield	Olfactive Description	Yield	Olfactive Description
Rose *(Rosa centifolia* L.)	0.47%	Fruity facets (banana), floral, dew, honey	0.22%	Green smell, floral, sticky, tenacious.
Lily (*Lilium* spp.)	0.41%	Floral scent, stifling, honey, less greasy	0.36%	Strong smell of lilies, greasy, green, aqueous and iodine.
Jasmine (*Jasminum grandiflorum* L.)	2.60%	Greasy odor, floral, jasmine and minty.	0.67%	Floral odor, sticky, greasy, animal and heady.

**Table 3 molecules-28-00891-t003:** Constituents identified in the essential oil from *C. wildii* resin. Data reported according to their retention index values (RI) and relative abundance defined as a surface percentage of the FID chromatogram (% FID) and with the corresponding standard deviation (SD). Cal. and lit. stand for calculated and literature, respectively.

Compounds	RI Cal. Apolar	RI Lit. Apolar	RI Cal. Polar	RI Lit. Polar	% FID Apolar	SD
Ethanol *	-	440	946	922	tr	-
Ethyl formate *	-	488	838	823	tr	-
Ethyl acetate *	-	599	903	876	tr	-
Acetic acid *	-	633	1473	1446	tr	-
Heptane	700	700	701	700	2.31	±0.03
Toluene	752	750	1055	1033	tr	-
Octane	799	800	801	800	tr	-
Isovaleric acid *	821	880	-	1655	tr	-
4-Heptanone *	850	853	1141	1146	tr	-
3-Heptanone	863	866	1170	1175	tr	-
2-Heptanone	866	867	1199	1173	tr	-
4-Heptanol	872	874	1296	1250	tr	-
3-Heptanol	878	877	1307	1270	tr	-
2-Heptanol	882	882	1332	1305	tr	-
2,5-Diethyltetrahydrofuran *	885	884	1053	1051	tr	-
Prenyl acetate *	-	899	1272	1244	tr	-
Nonane	900	900	901	900	0.53	-
5,5-Dimethyl-2(5H)-furanone *	909	910	-	1585	tr	-
Tricyclene	922	920	1016	1015	0.10	-
*α*-Thujene	925	925	1038	1028	4.21	±0.04
Benzaldehyde *	929	929	1556	1511	tr	-
*α*-Pinene	937	932	1035	1021	63.55	±0.04
*α*-Fenchene	-	940	1068	1067	tr	-
Camphene	946	946	1077	1071	0.68	-
Verbenene *	948	946	-	1123	tr	-
Thuja-2,4(10)-diene	949	945	1139	1122	0.19	-
1-Heptanol *	951	954	1471	1453	tr	-
Methylheptenone	963	963	1358	1340	tr	-
Sabinene	968	968	1135	1124	5.01	±0.01
*β*-Pinene	975	972	1123	1113	15.95	±0.04
6-Methyl-5-hepten-2-ol *	-	976	1478	1464	tr	-
2-Pentylfuran	978	978	1248	1230	tr	-
Octanal	980	982	1308	1288	tr	-
Myrcene	982	983	1176	1161	tr	-
*o*-Methyl anisol *	989	983	1435	1411	tr	-
*trans*-Anhydrolinalooloxide *	994	993	-	1253	tr	-
*α*-Phellandrene	998	998	1179	1170	tr	-
*n*-Decane *	-	1000	1000	1000	tr	-
*ᴪ*-Limonene	-	1004	1183	1183	tr	-
1,4-Cineol *	1005	1006	1192	1167	tr	-
*δ*-3-Carene *	1006	1006	1162	1145	tr	-
*α*-Terpinene	1010	1009	1194	1186	0.23	±0.01
Benzyl alcohol *	-	1010	1908	1862	tr	-
*p*-Cymene	1013	1014	1289	1269	2.10	±0.04
*p*-1-Menthene	1018	1004	1147	1150	tr	-
Eucalyptol	1021	1022	1224	1214	tr	-
*β*-Phellandrene *	1022	1021	1224	1207	tr	-
Limonene	1023	1024	1214	1196	1.22	±0.01
*o*-Cymene *	1026	1006	-	1260	tr	-
(*Z*)-*β*-Ocimene	1026	1024	1250	1235	tr	-
*o*-Cresol *	1034	1024	2031	2016	tr	-
5-Methylhexanoic acid *	1036	1038	-	1914	tr	-
(*E*)-*β*-Ocimene	1037	1037	1267	1250	0.18	-
*γ*-Terpinene	1050	1051	1263	1245	0.51	±0.01
4-Nonanone *	1053	1053	1344	1390	tr	-
1-Octanol *	1054	1055	1575	1547	tr	-
*trans*-Sabinene hydrate	1054	1058	1482	1465	tr	-
Heptanoic acid *	1066	1074	1975	1935	tr	-
3-Nonanone *	1066	1071	1375	1354	tr	-
2-Nonanone *	-	1071	1408	1392	tr	-
*cis*-Linalool oxide (furanoid) *	-	1062	1462	1441	tr	-
Fenchone *	-	1072	1423	1405	tr	-
*trans*-Linalool oxide (furanoid) *	1073	1076	1491	1463	tr	-
*p*-Cymenene	1074	1077	1461	1438	tr	-
Terpinolene	1080	1080	1301	1286	0.11	±0.01
Linalool	1084	1087	1562	1549	0.12	±0.02
*α*-Pinene oxide	1085	1085	1399	1364	tr	-
*cis*-Sabinene hydrate *	1086	1095	1570	1556	tr	-
Perillene *	1087	1063	1440	1425	tr	-
*α*-Thujone	1088	1088	1467	1431	tr	-
Heptyl Acetate *	1093	1095	-	1364	tr	
*β*-Thujone	1099	1096	1467	1443	tr	-
Rosefuran	-	1099	1421	1415	tr	
4,8-Epoxyterpinolene *	-	1099	1491	1477	tr	-
Chrysanthenone	1102	1102	1537	1510	tr	-
*α*-Campholenal	1106	1099	1517	1481	tr	-
Terpinen-1-ol *	-	1107	1596	1572	tr	-
Nopinone	1110	1107	1614	1565	tr	-
Octyl formate	1111	1112	-	1560	tr	-
*cis*-Verbenol	-	1111	1681	1660	tr	-
*cis-p*-Menth-2,8-dien-1-ol *	1117	1117	-	1610	tr	-
*allo*-Ocimene	1118	1117	1391	1367	tr	-
Camphor	1123	1123	1548	1517	tr	-
*trans*-*p*-Menth-2-en-1-ol *	1125	1104	1584	1566	tr	-
*trans*-Pinocarveol	1126	1128	1683	1654	0.38	-
Thujanol *	1126	1128	-	1622	tr	-
*trans*-Verbenol	1130	1129	1704	1680	0.32	±0.01
Sabina ketone	-	1133	1668	1624	tr	-
Pinocarvone	-	1138	1601	1567	tr	-
*trans*-Pinocamphone	1139	1146	-	1526	tr	-
*cis*-Pinocamphone	1141	1153	1578	1534	tr	-
Isoborneol *	-	1143	1695	1669	tr	-
*α*-Phellandren-8-ol	1147	1148	1750	1710	tr	-
Borneol *	1154	1154	1729	1702	tr	-
*p*-Methylacetophenone	1155	1154	-	1759	tr	-
*m*-Methylacetophenone *	-	1157	1812	1786	tr	-
*α*-Thujenal *	1159	1167	1660	1642	tr	-
*p*-Cymen-8-ol	1161	1165	1875	1849	tr	-
Myrtanal *	1161	-	-	1543	tr	-
Octanoic acid	1163	1175	2080	2048	tr	-
Terpinene-4-ol	1164	1164	1626	1600	0.64	-
*β*-Pinene oxide *	1167	1153	-	1366	tr	-
Myrtenal	1171	1171	1663	1624	tr	-
*α*-Terpineol	1174	1175	1721	1693	tr	-
*p*-Mentha-1,5-dien-7-ol	1177	1191	1827	1814	tr	-
Myrtenol	1180	1183	1821	1791	tr	-
Verbenone	1182	1182	1746	1723	tr	-
Decanal *	1184	1185	-	1492	tr	-
*cis*-Piperitol	1191	1188	-	1741	tr	-
Octyl acetate *	1192	1187	1493	1465	tr	-
*trans*-Piperitol *	1193	1187	-	1669	tr	-
*cis*-Carveol	1198	1205	1861	1861	tr	-
*n*-Dodecane *	1199	1200	-	1200	tr	-
Fenchyl acetate *	1208	1200	-	1477	tr	-
Bornyl formate *	1213	1190	-	1610	tr	-
Cuminaldehyde	1213	1217	1818	1774	tr	-
Carvone	1216	1217	1770	1732	tr	-
Isobornyl formate *	1219	1222	-	1596	tr	-
Carvotanacetone	1223	1220	1714	1665	tr	-
Piperitone	1229	1231	1764	1733	tr	-
Carvenone	1232	1226	1753	1737	tr	-
Perrilaldehyde *	1248	1248	-	1787	tr	-
Phellandral *	1252	1237	-	1727	tr	-
(*E*)-Anethole *	1262	1265	-	1827	tr	-
*p*-Cymen-7-ol *	1264	1278	2131	2101	tr	-
Thymol *	1268	1277	2215	2167	tr	-
Bornyl acetate	1270	1273	1606	1586	tr	-
Isobornyl acetate *	1271	1276	-	1582	tr	-
*cis*-Verbenyl acetate *	1275	1264	-	1655	tr	-
Carvacrol *	1277	1287	2197	2204	tr	-
Menthyl acetate *	1278	1280	-	1560	tr	-
Perillyl alcohol *	-	1281	2035	2001	tr	-
6-Hydroxycarvotanacetone *	-	1281	1898	-	tr	-
*trans*-Pinocarvyl acetate *	1281	1305	1679	1661	tr	-
4-Terpinenyl acetate *	1284	1284	1641	1640	tr	-
Myrtenyl acetate	1306	1304	1717	1685	tr	-
Syringol *	-	1307	2177	2272	tr	-
Piperitenone *	1311	1321	1965	1909	tr	-
*trans*-Carvyl acetate *	1315	1307	1761	1727	tr	-
Geranic acid *	1316	1348	-	2328	tr	-
*δ*-Elemene	-	1337	1487	1465	tr	-
*trans*-Sobrerol *	1346	1350	-	2338	tr	-
Decanoic acid *	-	1362	2293	2272	tr	-
*α*-Ylangene *	1378	1370	-	1468	tr	-
*β*-Elemene *	1390	1386	1611	1591	tr	-
Cuminyl acetate *	1391	1405	1998	1981	tr	-
Tetradecane *	1399	1400	-	1400	tr	-
*α*-Gurjunene *	1413	1403	-	1510	tr	-
*α*-Cedrene	-	1411	1633	1587	tr	-
*trans-β*-Caryophyllene *	1421	1420	-	1596	tr	-
Aromadendrene	1441	1446	1633	1615	tr	-
Alloaromadendrene *	1462	1461	-	1638	tr	-
2-Tridecanone *	1475	1478	-	1794	tr	-
*β*-Selinene *	1485	1478	-	1724	tr	-
Valencene *	-	1485	1750	1751	tr	-
Eremophyllene *	1485	1486	-	1743	tr	-
*β*-Himachalene *	1494	1469	-	1718	tr	-
Dihydro-*β*-agarofurane *	1500	1495	1755	1704	tr	-
*γ*-Cadinene	1509	1511	1788	1758	tr	-
*δ*-Cadinene *	1516	1516	1784	1755	tr	-
(*E*)-Nerolidol *	1548	1550	2056	2042	tr	-
Spathulenol	1568	1567	2155	2129	tr	-
*β*-Caryophyllene oxide *	1575	1569	2025	1977	tr	-
Globulol *	1579	1581	2107	2074	tr	-
Viridiflorol *	1586	1583	2123	2087	tr	-
Epi-*γ*-Eudesmol *	1612	1607	2148	2100	tr	-
Isospathulenol *	1628	1626	2290	2231	tr	-
*β*-Eudesmol *	1639	1633	2275	2234	tr	-
*α*-Eudesmol *	1658	1639	-	2188	tr	-
Octyl caprylate *	1760	1771	-	2020	tr	-
Cembrene	1933	1928	2214	2193	tr	-
Thunbergol *	2049	2032	-	2575	tr	-

* Identified thanks to fractionation.

**Table 4 molecules-28-00891-t004:** Antifungal activities of *C. wildii* expressed as diameter of inhibition (mm) and as MIC_80_ (µg/mL) on two Candida species.

Compound/Strains	Candida Albicans ATCC 1066	Candida Glabrata ATCC 90033
	Diameter of Inhibition (mm)	
Essential oil of *C. wildii*	16	22
Amphotericin B	30	30
	MIC 80 *	
Essential oil of *C. wildii*	2.7 mg/mL	2.7 mg/mL
Amphotericin B	0.5 µg/mL	0.5 µg/mL

Amphotericin B: Antifungal agent used as positive control. * Minimum inhibitory concentration: The minimum concentration that inhibits 80% of fungal growth.

**Table 5 molecules-28-00891-t005:** Antimicrobial activity expressed as MIC (mg/mL) of *C. wildii* essential oil on 21 Gram positive and negative bacteria strains.

Bacteria (Strains Numbers)	MIC (mg/mL)
*Cutibacterium acnes* (210–263, 210–762, 210–878, 210–982, 210–088)	>100
*Prevotella* sp. (215–843, 210–722, 215–425)	>100
*Methicillin-resistant Staphylococcus aureus* (210–468, 210–004)	30
*Methicillin-suscpetible Staphylococcus aureus* (215–254, 210–352, 215–044)	>100
*Streptococcus pyogenes* (215–140, 210–622, 210–951)	30
*Corynebacterium tuberculostearicum* (215–122, 215–378)	100
*Gemella morbillorum* (210–137)	>100
*Gemella haemolysans* (215–853)	>100
*Porphyromonas asaccharolytica* (210–873)	100

**Table 6 molecules-28-00891-t006:** Positive controls included in bioassays.

Assay	Positive Control
Hyaluronidase assay	Hydraberry commercial extract
Tyrosinase assay	SymWhite ingredient
DPPH radical scavenging assay	*Rosmarinus officinalis* L. commercial extract
Elastase assay	Berryflux vita (VITALAB)
Lipoxygenase assay	Resveratrol

## Data Availability

The data is stored in our laboratory.

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
