# Peer review of "Commiphora wildii Merxm. Essential Oil: Natural Heptane Source and Co-Product Valorization"

_molecules, 2023, doi:10.3390/molecules28020891_

Round 1

Reviewer 1 Report

Dear authors,

The manuscript entitled "Commiphora wildii Merxm. essential oil: natural heptane source and co-product valorization” describes twofold, first evaluate heptane from C. wildii essential oil as a raw material extraction solvent, second; determine the composition of C. wildii essential oil after heptane isolation.  It presents scientific relevance for the Chemistry.

However, you need to change some details/information in the abstract, Introduction, Material and Methods, Results, discussion and “conclusions”.

-line 32: variousways

-line 54-58: the sentences are too long, the topic is lost

-line 79: C. wildii -it is written in italics. attention throughout the manuscript

- table 2: please note the database from where the Olfactive description was noted

- please note below the table 3 the abbreviations used

-please improve the quality of the figure 2

-line 167: mL

Conclusion: Adequate, but I suggest to indicate disadvantages/limitations of the method and the study! Perhaps, to highlight the text in the 'Limits of the study' section (extend the conclusions).

References: Spelling of references must be checked to meet the journal style.

Reviewer 2 Report

Summary of the key contribution of the paper:

The manuscript of Commiphora wildii Merxm. Essential Oil: Natural Heptane Source and Co-Product Valorizationinvestigation on essential oil of Commiphora wildii Merxm oleo gum resin as a source of heptane is reported here. Heptane, representing up to 30 wt-% of this oleo gum resin up to a value of 30 wt-% was successfully isolated from the C. wildii essential oil using an innovative double distillation process. Isolated heptane was then used as a solvent to extract some noble plants of perfumery. In addition, their investigation on the composition of the remaining essential oil after heptane isolation revealed ten compounds were found in majority in the essential oil, a-pinene being the most abundant compound. The scope of the manuscript is very interesting, the manuscript can be accepted.

Highlights:

·        This article clearly articulates the first time C. wildii was used as a source of heptane naturally present in its essential oil. The latter was found to contain up to 30 % heptane. A simple double distillation process was proposed, yielding heptane of high purity, containing 0.3% alpha-pinene and 0.1 % alpha-pinene.

·        The figures and tables are well referenced and clear.

·        The resulting essential oil after removal of heptane exhibited important hyaluronidase and lipoxygenase activities, but low elastase activity. These activities suggest possible interesting antiaging and skin repairing cosmetics properties. These properties seem to conform to the traditional usage of the resin of C. wildii.

·        The tests against a several strains of bacteria and two Candida species revealed inhibition preferentially on pathogenic and not commensal human bacterial strains of the skin or oral cavity and a medium effect against C. albicans and a potent inhibitor of C. glabrata.

·        This study shows different biological properties for the doubly distillated essential oil of C. wildii, which could be valorized as a raw material for cosmetic usage, after the removal of the natural heptane for perfumery usage.

·        Lowlights:

·        There are no Lowlights in this paper.

Author Response

We thank the reviewer for his very positive opinion of the manuscript.